# Enhancing Dental Applications: A Novel Approach for Incorporating Bioactive Substances into Textile Threads

**DOI:** 10.3390/pharmaceutics15102487

**Published:** 2023-10-18

**Authors:** Marek Pokorný, Jolana Kubíčková, Jan Klemeš, Tomáš Medek, Adam Brýdl, Martina Pachovská, Tereza Hanová, Josef Chmelař, Vladimír Velebný

**Affiliations:** R&D Department, Contipro a.s., 56102 Dolní Dobrouč, Czech Republic; jolana.kubickova@contipro.com (J.K.); jan.klemes@contipro.com (J.K.); tomas.medek@contipro.com (T.M.); adam.brydl@contipro.com (A.B.); martina.kralova@contipro.com (M.P.); tereza.hanova@contipro.com (T.H.); josef.chmelar@contipro.com (J.C.); vladimir.velebny@contipro.com (V.V.)

**Keywords:** hyaluronic acid, octenidine dihydrochloride, threads, coating of active substances

## Abstract

In the realm of surgical and dental applications, hyaluronic acid (HA) braided threads show significant therapeutic potential due to their incorporation of pharmaceutical active ingredients. This study primarily focuses on resolving the crucial challenge of devising a deposition method that can ensure both precision and uniformity in the content of the active ingredient Octenidine dihydrochloride (OCT) within each segment of the threads. Our objective in this study was to develop a continuous deposition method for OCT onto a braided thread composed of 24 hyaluronic acid-based fibers, aiming for a specific OCT content of 0.125 µg/mm, while maintaining a maximum allowable deviation of ±15% in OCT content. The motivation behind designing this novel method stemmed from the necessity of employing a volatile solvent for the active agent. Conventional wetting methods proved unsuitable due to fluctuations in the solution’s concentration during deposition, and alternative methods known to us demanded intricate technical implementations. The newly introduced method offers distinct advantages, including its online processing speed, scalability potential, and cost-efficiency of the active agent solution. Additionally, it minimizes the impact on the natural polymer thread, preserving energy by obviating the need for complete thread saturation. Our research and precise apparatus development resulted in achieving the desired thread properties, with an OCT content of (1.51 ± 0.09) µg per 12 mm thread piece. These findings not only validate the suitability of this innovative method for depositing active agents but also extend its potential applicability beyond dental care.

## 1. Introduction

Linear textile formations, imbued with biologically active substances, play crucial roles in contemporary medicine. They find utility in diverse applications, such as surgical sutures [1,2], interdental cleaning floss [3], reinforcing scaffolds [4,5], and a multitude of other domains [6]. Several methodologies exist for incorporating these active substances into the fabric’s structure [7].

Among the simplest techniques is immersing or dipping the textile into a solution containing the active substance. During this process, the substance gradually adheres to the surface of the textile [2,8,9]. Post immersion, excess solution removal can be accomplished through mechanical squeezing [10] or drying procedures [11]. Alternatively, more sophisticated methods are available, including non-contact deposition techniques like drop coating [12], spray coating [13,14,15], ink-jet coating or printing [16,17], coaxial nozzle coating [18], plasma and ionized gas deposition [19], or electrostatic spinning [20,21,22]. Furthermore, active agents can be introduced into the textile yarns through contact deposition, employing coating principles [23].

Hyaluronan (HA) is a linear polysaccharide that is naturally present in the human body. The monomer unit of HA consists of D-glucuronic acid and N-acetyl-D-glucosamine, which are linked by alternating β(1 → 4) and β(1 → 3) glycosidic bonds. HA plays an important role in many biological processes including wound healing and tissue regeneration, hydration and lubrication. Due to its properties, HA is widely used in medicine, including wound healing applications [24,25,26].

Hyaluronan fibers and their braided configurations are particularly attractive for use in dental and surgical domains. The distinctive structure and composition of these braided filaments are safeguarded by Czech patent CZ308980 [27], which emphasizes the use of filaments containing OCT as an active ingredient. Ensuring the precise and stable deposition of OCT became paramount, not only to meet the stringent requirements for active substance content in 12 mm thread segments but also to validate its suitability for pharmaceutical production. To our knowledge, such a method was previously non-existent and necessitated novel development.

Octenidine dihydrochloride (OCT) is a cationic surfactant, an antiseptic containing two pyridine nuclei linked by an aliphatic chain. Its most common form is the dihydrochloride salt. At physiological pH, it is fully ionized (carries a positive charge), is stable over a wide pH range (1.6 to 12.2), and exhibits resistance to UV exposure. Due to its positive charge, octenidine interacts with negatively charged structures on the surface of micro-organisms (in particular, the lipid components of bacterial cell walls), thereby disrupting the function of their plasma membranes and causing autolysis of these cells. It is therefore characterized by high antimicrobial activity without disrupting the cell epithelium of the healing wound. The cationic nature of octenidine minimizes its absorption by the skin or mucosa. OCT is a registered antiseptic with cutaneous and mucosal application. OCT itself is not toxic and does not cause allergic reactions [25,26].

The primary impetus behind this development stemmed from the imperative for employing volatile solvent of the active agent, a context in which conventional wetting methods proved unsuitable. In typical wetting processes, the solution’s concentration tends to increase, leading to alterations in the active ingredient content of the thread. Recognizing the stringent tolerance limits set for permissible variations in active ingredient content for product registration, we prioritized the precision and long-term stability of the coating process. Other methods, as previously mentioned, necessitate sophisticated technical equipment, and their ability to consistently meet our criteria for active substance content has not been adequately demonstrated by these methods.

These compelling factors underscored the necessity to devise a novel coating method that met specific criteria. This method was envisioned to be straightforward, robust, easily scalable to larger production volumes, and ideally conducive to minimizing waste in the utilization of the active ingredient solution.

In this study, we drew inspiration from the realm of contact deposition techniques to develop an innovative approach for applying active agents to linear textile formations. By adapting principles from general surface treatment procedures, we aim to advance the field of textile-based biologically active substances, expanding their potential across various applications [7,28].

## 2. Theories and Hypotheses

This study presents an innovative approach for precisely depositing Octenidine dihydrochloride (OCT) as an active ingredient onto spliced yarns comprised of hyaluronic acid (HA)-based fibers. Our method leverages roll-to-roll yarn rewinding combined with a continuous coating technique that incorporates principal elements of both kiss coating [7] and drop coating [12].

In this process, the textile thread comes into contact with the external wall of a dispensing nozzle during rewinding. Notably, the solution rises up the wall of the vertically mounted conical nozzle without dripping. Despite the apparent simplicity of this principle, the development of this method presented several intricate technical challenges.

The coating apparatus underwent optimization to ensure a consistent thread rewind rate, maintain the mechanical stability of the thread in relation to the nozzle’s position, and sustain the temporal stability of solvent evaporation from the OCT solution on the nozzle surface. Furthermore, we had to stabilize the mechanical properties of the yarn before deposition, with particular emphasis on preserving its ductility. Any fluctuations in ductility during deposition had the potential to disrupt the desired rewind speed and impact the yarn’s dimensional parameters during wetting.

The meticulous evaluation of highly critical process parameters was carried out systematically to consistently achieve the desired OCT content within the defined tolerance, even on shorter segments of the yarn. Overcoming these challenges was imperative for the successful implementation of our novel deposition method.

The application of the active substance onto the textile thread occurs precisely as the thread makes contact with the nozzle wall. During the rewinding process, the nozzle dispenses the solution containing the active agent, harnessing the capillary effect as the solution ascends the external walls of the vertically positioned conical nozzle, as depicted in Figure 1.

The coating principle hinges on the consistent and even absorption of the solution throughout the winding of the textile yarn. This absorption occurs on the surface of the fibers and within the capillary spaces that exist between them. It is essential for the volume of the yarn’s pores, into which the solution permeates, to remain constant and unchanging throughout the process. Technically, these variable properties cannot be influenced in any way; the apparatus merely ensures a constant and time-steady dosing of the solution.

Nevertheless, it is noteworthy that a portion of the solution, which is not absorbed into the yarn’s structure, evaporates from the nozzle surface. Consequently, the rate of evaporation must remain stable over time, even though the thread can potentially absorb a significantly larger quantity of the dispensed solution. In essence, there must exist an equilibrium between the volume absorbed and the discrepancy between the administered and evaporated solution volumes.

This equilibrium can be succinctly expressed through a simple equation:(1)AR=FR−ER
where *AR* is the absorption rate, *FR* is the feed rate and *ER* is the evaporation rate. To maintain the equilibrium described in the previous section, the following conditions must be meticulously adhered to:(a)The solution feed rate emanating from the nozzle must remain unwaveringly constant. The dosing rate is directly correlated with the ultimate active substance content of the textile yarn.(b)The concentration of the active substance within the dosing solution must remain unwavering and homogeneous, ensuring that the active substance is uniformly distributed throughout the solution. The concentration is directly tied to the final active substance content of the textile thread.(c)The speed at which the thread is rewound must remain constant. This rewinding speed holds an inverse relationship with the final active substance content of the textile thread, as articulated in Equation (1). In practical terms, this signifies that every thread segment, when in contact with the nozzle, must absorb the solution with the active substance for the same duration. Consequently, thread rewinding was meticulously regulated through constant-speed motors, with an optical encoder employed for control and recording the rewinding speed waveform during deposition.(d)The resulting thread segment’s length must remain consistent. This length is directly proportional to the final active substance content of the textile thread. In essence, altering the length of the thread segment would correspondingly affect the active substance content in proportion to the variation in length. To ensure uniformity, threads were precisely cut to the desired length using automated knives equipped with a thread feeder.

These crucial conditions can be concisely represented in the following equation:(2)$$ASC=\frac{CO\cdot FR\cdot TL}{RS}$$
where *ASC* (g) is the active substance content, *CO* (g/L) is the concentration of the active substance in the dosing solution, *FR* (L/h) is the feed rate, *TL* (m) is the length of the textile thread fragment, and *RS* (m/h) is the rewind speed. 

From this equation, it is possible to directly calculate the concentration of the active substance within a specified length of textile thread, provided that the process variables described above are set and the condition outlined in Equation (1) is met. With this fundamental equation in hand, it becomes feasible to assess the potential range of variation in the resulting active substance concentration. This range is governed by the principles of error propagation, as expressed in a general form in Equation (3) [29], which can then be further refined to compute the error associated with the function described in Equations (1)–(4).
(3)$$\sigma y^{2}=\overset{n}{\underset{i=1}{\sum }}{\left(\frac{\partial f}{\partial x_{i}}\right)}^{2}\cdot \sigma_{i}^{2}$$
(4)$$\sigma {ASC}^{2}={\left(\frac{FR\cdot TL}{RS}\right)}^{2}\cdot \sigma {CO}^{2}+{\left(\frac{CO\cdot TL}{RS}\right)}^{2}\cdot \sigma {FR}^{2}+{\left(\frac{CO\cdot FR}{RS}\right)}^{2}\cdot \sigma {TL}^{2}+{\left(-\frac{CO\cdot FR\cdot TL}{{RS}^{2}}\right)}^{2}\cdot \sigma {RS}^{2}$$
where the symbol *σ* denotes the error of the quantity.

The mathematical model described by Equation (2) does not encompass all the variables that can influence the final outcome. Consequently, it is necessary to consider additional prerequisites:(e)Maintaining a constant tensile force during the winding process is imperative to ensure uniform stretching of the thread, assuming consistent mechanical properties across its constituent sections or fragments. To achieve this, an electromagnetic device, known as a dancer, was incorporated into the coating apparatus to sustain a consistent tension force during thread rewinding.(f)Ensuring minimal relative changes in the length of thread sections is essential, as this aspect is linked to the dynamics of the entire thread rewinding process and the fluctuating relative tensile strength along the non-ideal thread’s length. To mitigate this, the braided thread was mechanically tightened before application to limit further stretching during rewinding. Mechanical stability in the coating apparatus was achieved through the utilization of rigid joints and anti-vibration components.(g)Maintaining a constant volume of the solution for wetting the thread fragments is crucial. The volume of the solution formed on the walls of the dispensing nozzle, where the thread is wetted directly correlates with the resulting active substance content. This volume is influenced not only by the dosing rate but also by the evaporation rate (as indicated by the equilibrium described in Equation (1)). Consequently, a constant evaporation rate of the solvent solution with the active substance is required. This evaporation rate is inversely proportional to the resulting active substance content. Such a condition necessitates the maintenance of a constant temperature within the dispensing nozzle and its surroundings, with no variations in air turbulence during the application process.(h)The length of the circular arc in contact with the rewound thread on the conical nozzle must remain constant (refer to Figure 1B). The extent of this arc, where contact and saturation of the thread on the nozzle’s surface take place, directly influences the resulting active substance content. Consequently, it necessitates the consistent positioning of the rewound thread’s height relative to the vertically clamped dispensing nozzle.(i)The textile thread should exhibit uniform absorbency along its entire length and maintain a consistent circular shape. Any deviations in shape or variations in the area of contact with the dispensing nozzle’s wall or the volume of solution on the nozzle’s surface would yield differing absorption values for the solution, resulting in varying active substance content along the thread’s length. The braided filament employed in our study lacks axisymmetry and does not possess a circular cross-section in the perpendicular plane (as depicted in Section 4). Furthermore, it is composed of filaments with varying properties. Therefore, any rotation of the filament relative to the nozzle’s wall could alter its local ability to absorb the active agent solution.

## 3. Materials and Methods

### 3.1. Materials

Native hyaluronic acid (Mw 360 kDa) and lauroyl derivative of hyaluronic acid (Mw 310–370 kDa) served as the primary materials for the braided threads, both of which were produced and provided by Contipro a.s. (Dolní Dobrouč, Czech Republic). The lauroyl hyaluronic acid derivative was prepared by esterifying the −OH groups on the HA backbone using a symmetric lauric acid anhydride [25], resulting in a final degree of substitution of 29–42%. Octenidine dihydrochloride (OCT), an active pharmaceutical ingredient, was supplied by Ferak (Berlin, Germany). Ethanol of 96% purity was used as a solvent to form a solution with the active ingredient, and it was supplied by Lach-Ner s.r.o (Neratovice, Czech Republic).

### 3.2. Thread Preparation

Fibers from hyaluronic acid and lauroyl hyaluronan derivative were prepared using the wet spinning process, as described in more detail in our previous publications [5,25]. 

In short, polymer HA was dissolved (HA in water, lauroyl HA in 50% 2-propanol), and the final concentration of the polymers was 50 mg·mL^−1^. The polymer solution was extruded into a coagulation bath at a flow rate of 200 µL·min^−1^. The coagulation bath contained a mixture of 2-propanol and lactic acid (volume ratio 80:20).

After preparation, the fibers are washed to remove residual coagulation bath residues and further processed through textile operations: winding, plying, and braiding into thread form.

The winding process involves the fibers, which are then collected in a cross-wound package. A Steeger spinning machine (FDSM, Wuppertal, Germany) was used for spinning the prepared fibers. To twist the fibers, a ring plying machine (VÚB a.s., custom-made machine, Ústí and Orlicí, Czech Republic) was used. Three fibers were twisted together at a spindle speed of 4800 rpm, and the material feed rate was set at 12 m·min^−1^. Braiding is a mechanical process where bobbins with yarn are guided along intersecting paths. These intersections interlace the fibers into a compact thread. The threads were braided using a horizontal machine (Steeger HS 80/48-15 VEA, Wuppertal, Germany). The number of crossings per centimeter determines the weaving density, which in our case was 22 pix·cm^−1^, and it depends on the ratio of the thread take-up speed and the braiding speed (the speed of the rotating head).

The stabilization process is carried out using the coating device, as depicted in Figure 2. This process involves the gradual elongation of the thread’s entire length during rewinding, while maintaining a constant tension force of 500 mN through the use of a dancer mechanism. Through this method, the cross-sectional structure of the thread, as shown in the diagram in Figure 3, is condensed and individual filaments are stretched, leading to overall thread elongation.

### 3.3. Coating of API

The braided thread, measuring 30 m in length and wound on a DIN80 textile spool, was threaded through the apparatus designed for the application of the active substance. It was securely clamped to the second spool, which facilitated the winding of the thread. To monitor the thread’s position on the nozzle wall, a camera (Andonstar, A1 2MP 500X USB, Shenzhen, China) was employed. In order to expedite solvent evaporation from the filament, a 500 mm-long infrared heating panel was utilized, with a surface temperature maintained at (60 ± 5) °C.

The active nozzle heater, featuring PID control, was set to a temperature of (32 ± 1) °C. The prepared solution, with an active agent concentration of 2.95 mg·mL^−1^, was dispensed (Chemyx, Nexus 6000, Stafford, TX, USA) into the conical-shaped coating nozzle (gauge 24) at a rate of 100 µL·min^−1^ during the thread rewinding process. The thread rewinding speed was configured at 2400 mm·min^−1^, while the electromagnetic linear dancer (Supertek, EDL 60, Emden, Germany) was adjusted to maintain a constant tension force of 500 mN.

The thread winding process was carefully controllId, incorporating cyclic movements to create a cross-winding pattern on the winding spool, with an angle of 60° between each thread. Following the application of the active agent, the thread was precisely cut into final 12 mm-length segments utilizing an automatic knife divider (JEMA JM-110LR, Taizhou, China). Concurrently, error estimates for all critical parameters were derived from Equation (2) and are presented in Table 1. The values from Table 1, along with the specified process parameters, were subsequently integrated into Equation (4).

Through the substitution of the values and estimated errors for the critical process parameters into Equations (2) and (3), the mean active ingredient content was determined with an absolute error of (1.48 ± 0.05) µg for a single thread segment. This theoretical calculation underscores that meticulous control of the critical process parameters within the specified tolerances (as outlined in Table 1) can lead to achieving an active ingredient content variation of less than 3.5%. It is important to note that this calculation primarily considers the principal critical process parameters and does not account for other factors influencing the coating process, as discussed in items (e) to (i) of Section 2.

### 3.4. Characterization of Thread Properties and API Content

Thread fineness was determined by weighing a 1 m length of thread on a Mettler Toledo balance (XSE 205, Columbus, OH, USA). The fineness was determined by the calculation: fineness (tex) = weight (g)/100 (m). Thread morphology and thread cross-section were visualized on a Zeiss (Ultra Plus, Oberkochen, Germany). Samples were cut with a special razor blade, coated with a layer of gold/palladium (8 nm) in a sputter coater (Leica, EM ACE600, Morrisville, NC, USA) and then imaged at 6.5 kV with SE2 and InLens detectors. Mechanical properties of the threads were measured using a 3343 single column testing system with a 100 N head (Instron, Norwood, MA, USA). The measurement was carried out at (23 ± 2) °C and at a relative humidity of (50 ± 5) %. The threads were pre-tensioned at 0.01 N and then stretched at a rate of 10 mm·min^−1^ until the value of 3 N was reached. The stretching of the thread was repeated ten times. 

For the analysis of active substance content, we selected 25 segments, each measuring 12 mm in length, from each 30 m-long thread, according to the following procedure. No segments were taken from the edges of the initial 3 m lengths. Within the remaining 24 m, we randomly selected 5 sections, from which we consecutively chose 5 segments, ensuring that one segment was always omitted among those selected. This method resulted in the analysis of a total of 25 segments from each thread. We prepared a total of 5 threads for this study. The selection process is depicted in Figure 4.

The OCT content of each selected thread segment was subjected to analysis. Octenidine was quantified in sample solutions following alkaline hydrolysis, utilizing high-performance liquid chromatography (HPLC) with reversed-phase separation and UV detection. Thread samples were dissolved to a concentration of 2 mg·mL^−1^ in solutions containing 50 wt.% EtOH (ethanol) and 0.5 wt.% NaOH (sodium hydroxide). The HPLC analysis was conducted using a Luna 5 µm C18(2) 100A column (250 × 4.6 mm). The mobile phase flow rate was set at 1.2 mL·min^−1^, the column temperature was maintained at 45 °C, the detector wavelength was set to 280 nm, and the injection volume was 15 µL. Between each two samples, a blank injection was introduced to cleanse the column and validate the stability of the baseline. All samples were analyzed in duplicate, and the results were averaged. The octenidine content in the threads was subsequently expressed as µg OCT per 12 mm segment, rounded to two decimal places.

## 4. Results and Discussions

The microscopic structure of the spliced 24-filament HA yarn with a crossover of 22 pix·cm^−1^ can be seen in the example sections in Figure 5. The fineness of the filaments was determined by weighing and recalculation to be (260 ± 40) tex.

The yarns exhibited a measured breaking strength of (8 ± 2) N and an elongation at break of (25 ± 5) %. It is important to note that these values, while indicative of yarn properties, primarily pertain to yarn processing considerations (e.g., active substance application, rewinding, cutting, etc.), as the intended application of the yarns does not subject them to significant mechanical stress. To assess the impact of gradual retraction of the cross-structure of the yarns and the inherent ductility of the yarns themselves on the rewinding rate during active agent application, we subjected the yarns to characterization under cyclic uniaxial force loading conditions. The threads were subjected to cyclic loading, reaching a maximum load of 3.0 N (approximately one-third of the thread’s breaking strength), while monitoring changes in ductility and relative length elongation, as depicted in Figure 6.

The graph in Figure 6 illustrates a gradual decrease in ductility (thread stretching) after eleven cycles of loading, indicating a partial stabilization of the mechanical properties of the thread under force loading conditions. Consequently, to prepare each thread for deposition and ensure consistent OCT active ingredient content results, it underwent rewinding four-to-eight times at a force of 500 mN. The number of repetitions was determined based on the criterion that the measured variation in rewinding rate fell below 3%, as indicated by the error estimate in Table 1.

The directly measured OCT masses from each of the five filaments are depicted in Figure 7. To enhance clarity, all data were processed into a box plot presented in Figure 8, which also includes other statistical variables (numerical average values). The resultant values have been further summarized in Table 2. 

In all cases, the results for the active substance content remain comfortably within acceptable limits, with a substantial margin of tolerance. The equilibrated mean OCT content values serve as strong indicators of the stability of both the coating and the process over five repetitions. The consistently low standard deviations provide further confirmation of the precision and accuracy achieved through the newly developed method for OCT deposition on spliced HA fiber yarn.

The mean value of the active substance content is (1.51 ± 0.09) µg/piece out of 125 pieces analyzed in five replicates, which corresponds to a relative error of 5.8%.

## 5. Conclusions

This paper presents the results obtained through the utilization of a newly developed method for the application of an active pharmaceutical ingredient to a braided thread composed of 24 fibers featuring two different modifications of hyaluronic acid. Octenidine was chosen as the selected API, with the target content in 12 mm segments of thread falling within the window ranging from 1.25 to 2.00 µg/piece. By harnessing a comprehensive understanding of the process and meticulously fine-tuning the experimental apparatus for API application, we achieved consistent and precise deposition of the active substance within the desired average range and well within the specified tolerance. The tolerance range was established at ±15% of the mean value of 1.50 µg/piece.

The theoretical OCT content, ascertained as (1.48 ± 0.05) µg/piece through the derived equation by fitting process variable values and estimating their errors, was used as the benchmark. OCT content was analyzed across randomly selected sections of five threads spanning 30 m lengths, totaling 125 measurements. The results prominently demonstrate that only one analyzed OCT content value approaches the tolerance limit, while the remaining results collectively validate the precision of the novel coating method. These findings culminate in an average OCT content of (1.51 ± 0.09) µg/piece, reflecting a relative deviation of 5.8%.

This method of API deposition showcases numerous advantages, including its economical use of API solution, gentle treatment of threads due to the application of low pulling forces, and minimal energy requirements for the evaporation of residual solvents from the thread. The newly developed technique for the precise application of API to braided hyaluronan yarns has significantly enhanced the functional properties of the yarn, rendering it suitable for a broad array of pharmaceutical applications.

## 6. Patents

This article is covered by the granted patent CZ308980 and its other national applications.

## Figures and Tables

**Figure 1 pharmaceutics-15-02487-f001:**
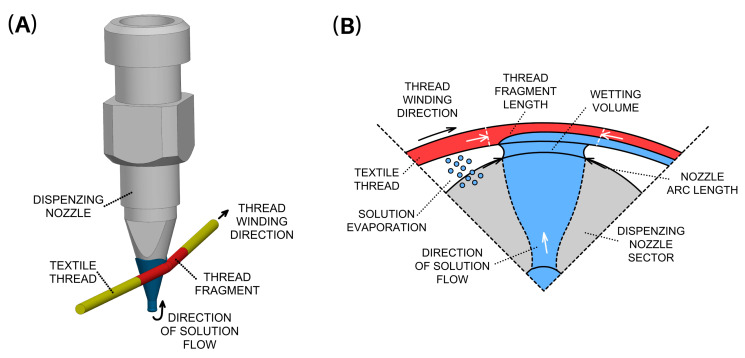
Schematic representation of the active agent application process on a rewound textile thread: (**A**) side view of the nozzle with the thread, (**B**) bottom view of the nozzle cut-out, highlighting key parameters.

**Figure 2 pharmaceutics-15-02487-f002:**
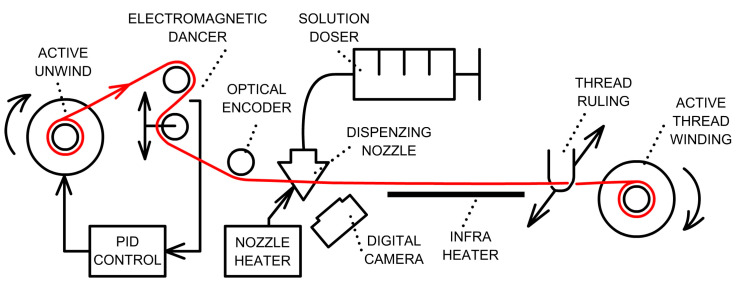
Comprehensive schematic of the apparatus for applying active agents to textile threads.

**Figure 3 pharmaceutics-15-02487-f003:**
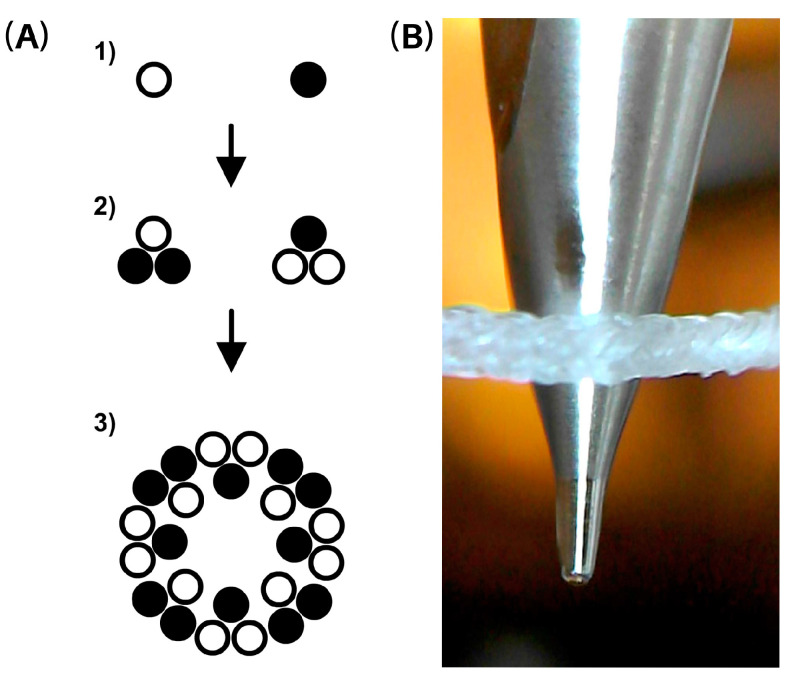
(**A**) Thread preparation procedure, illustrated in thread and fiber cut: (**1**) single HA and lauroyl HA derivative fibers, (**2**) three twisted fibers, and (**3**) braided thread. (**B**) Snapshot of rewound thread with API dispensing nozzle.

**Figure 4 pharmaceutics-15-02487-f004:**
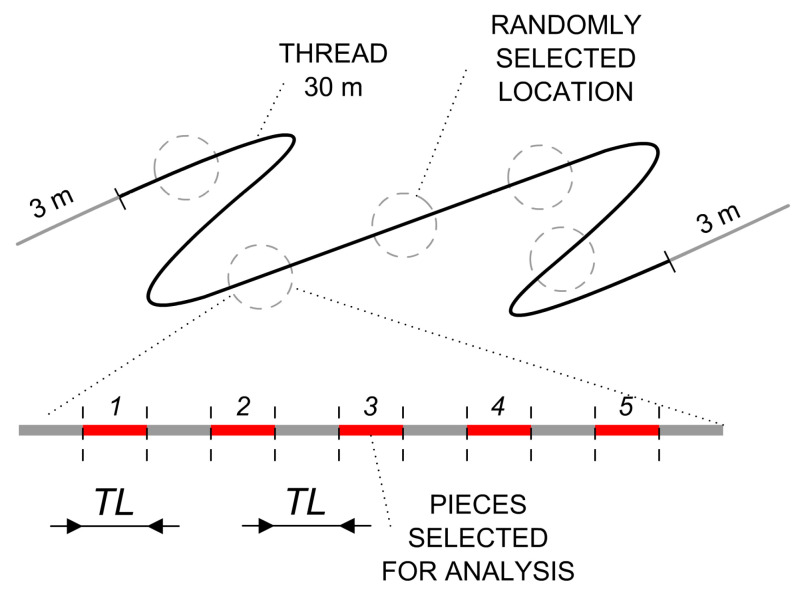
An illustrative representation elucidating the process of selecting thread segments for the analysis of active substance content. Five segments (1–5), highlighted in red, are chosen for analysis from a single section.

**Figure 5 pharmaceutics-15-02487-f005:**
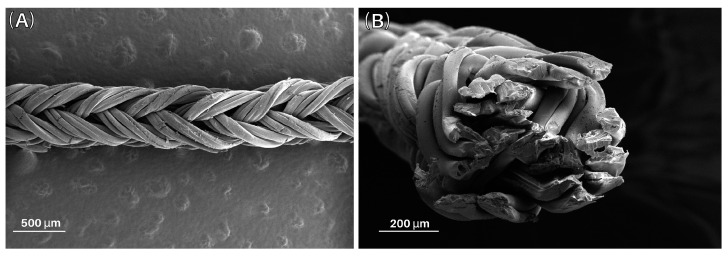
Images depicting the microstructure of the spliced yarn, offering insights into both the lateral direction (**A**) and the cross-sectional view (**B**).

**Figure 6 pharmaceutics-15-02487-f006:**
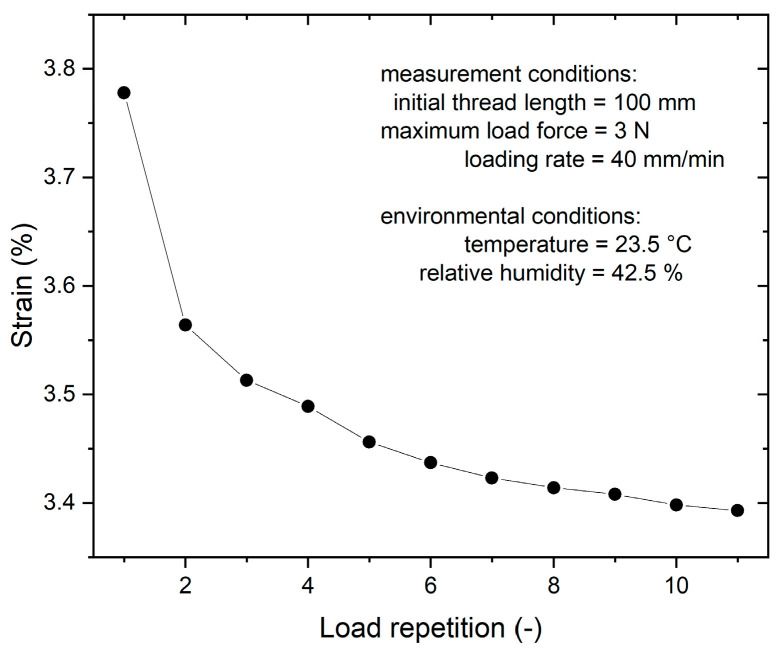
Dependence of mechanical deformation under cyclic loading of the thread up to 3.0 N.

**Figure 7 pharmaceutics-15-02487-f007:**
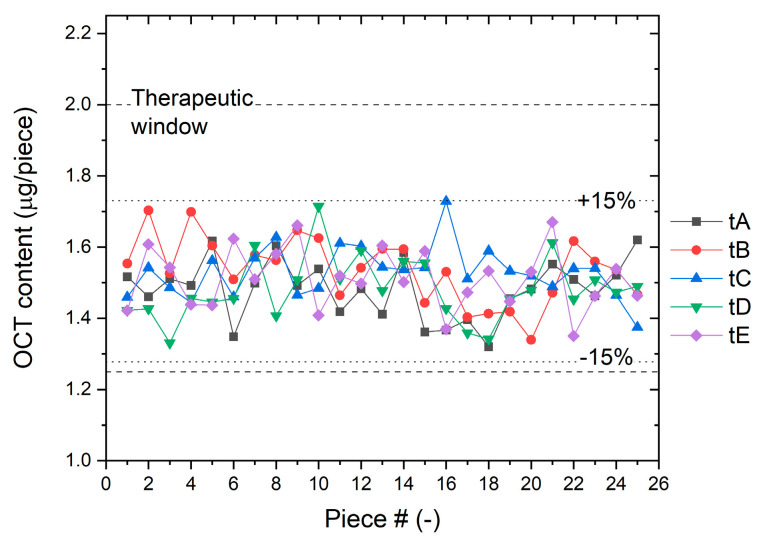
The results of OCT active substance mass obtained from five threads, with 25 pieces selected from each thread. The dashed lines in the graph delineate the tolerated range of values.

**Figure 8 pharmaceutics-15-02487-f008:**
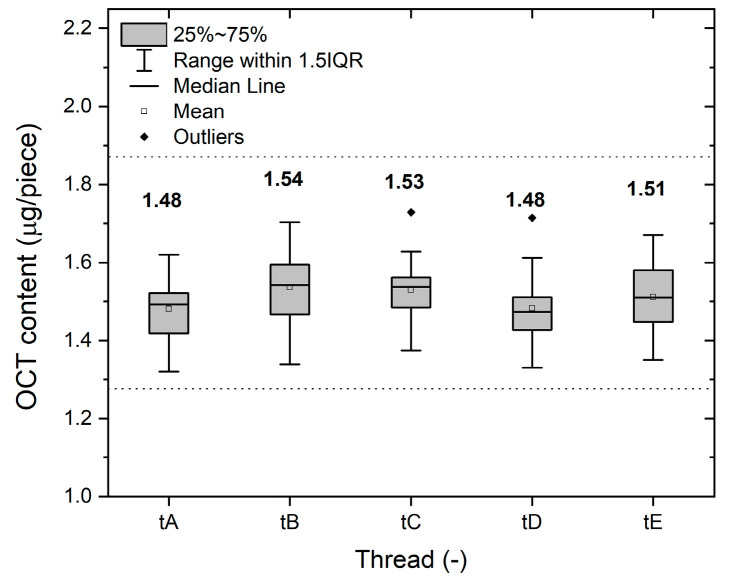
The results of OCT active substance mass obtained from five threads. The dashed lines show the tolerated range of values. Plotted data show the mean value.

**Table 1 pharmaceutics-15-02487-t001:** Controlled process variables from Equation (2) and estimations of their absolute and relative deviations.

Quantity	Setpoint	Absolute Error	Relative Error
Feed rate	100 µL·min^−1^	1 µL·min^−1^	1%
Solution concentration	2.95 mg·mL^−1^	0.03 mg·mL^−1^	1%
Rewind speed	2400 mm·min^−1^	72 mm·min^−1^	3%
Thread length	12.00 mm	0.12 mm	1%

**Table 2 pharmaceutics-15-02487-t002:** Statistical variables and OCT active ingredient content values obtained from five threads and 25 pieces selected from each thread.

Thread	Mean Value [µg/Piece]	Standard Deviation [µg/Piece]	Relative Error [%]	Minimum [µg/Piece]	Maximum [µg/Piece]
tA	1.481	0.083	5.60	1.320	1.620
tB	1.536	0.092	6.01	1.340	1.703
tC	1.529	0.072	4.69	1.374	1.728
tD	1.483	0.089	5.98	1.331	1.714
tE	1.511	0.086	5.69	1.350	1.670

## Data Availability

All data are published in the text of this publication and there are no further links to any data.

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
