# Peer review of "Enhancing Dental Applications: A Novel Approach for Incorporating Bioactive Substances into Textile Threads"

_pharmaceutics, 2023, doi:10.3390/pharmaceutics15102487_

Round 1

Reviewer 1 Report

The abstract requires improvement by adding more explanation regarding the purpose of the study, methods, and results. It should be more informative.

The methodology was not clearly stated. The authors have to tune it up, especially on the methodology, significance of the study, and the discussion of the results.  Some specific comments highlighting areas of the strength and weakness of this paper as below: The structure of this manuscript needs to be improved.

The introduction needs to be improved significantly. The introduction should focus on the review of previous work and the significance of your work. The methods are explained in the Introduction part make it confusing. In the introduction, the topic, problem statement, and research gaps should be explained more in detail.

The methodology was not clearly stated. The authors have to tune it up, especially on the methodology, significance of the study, and the discussion of the results.

Minor editing of English language required

Author Response

Assigned Editor:          Krongkarn Watakulsin

Journal:                       Pharmaceutics

Manuscript ID:             pharmaceutics-2644409

Type:                           Article

Section:                       Biologics and Biosimilars

Special Issue:              Hyaluronic Acid for Medical Applications

Submission Received:  18 September 2023

Title:

Enhancing Dental Applications: A Novel Approach for Incorporating

Bioactive Substances into Textile Threads

First, let me thank the reviewers for their valuable and substantive comments, which are greatly appreciated by the authors of this manuscript.

Text modifications are highlighted with colors as follows:

Green – new text,

Yellow – removed text,

Gray – replaced text.

Reviewer 1

Comments and Suggestions for Authors

Reviewer: The abstract requires improvement by adding more explanation regarding the purpose of the study, methods, and results. It should be more informative.

Answer: Thank you for this recommendation. The abstract has been revised according to this comment.

Reviewer: The methodology was not clearly stated. The authors have to tune it up, especially on the methodology, significance of the study, and the discussion of the results.  Some specific comments highlighting areas of the strength and weakness of this paper as below: The structure of this manuscript needs to be improved.

Answer: Thank you for this comment. The authors believe that significant improvements have been made in the editing of the manuscript and that these general requirements have been met. The abstract has been significantly revised, the Introduction section has been modified, paragraph 3.1 Materials has been added and a section on thread preparation has been extended.

Reviewer: The introduction needs to be improved significantly. The introduction should focus on the review of previous work and the significance of your work. The methods are explained in the Introduction part make it confusing. In the introduction, the topic, problem statement, and research gaps should be explained more in detail.

Answer: Thank you for this recommendation. The introduction part has been significantly added to better contextualize the whole sense of novel method development.

Reviewer: The methodology was not clearly stated. The authors have to tune it up, especially on the methodology, significance of the study, and the discussion of the results.

Answer: Thank you for this comment, the substance of which is already contained in previous comments. Therefore, here too we believe that the manuscript has been sufficiently improved in this aspect.

Comments on the Quality of English Language

Reviewer: Minor editing of English language required.

Answer: The authors have gone through the whole text again and corrected all formal errors.

Sincerely yours,

Marek Pokorny

First and corresponding author

Marek Pokorny, Ph.D.

Senior Researcher / Head of the Group

Group of R&D Technologies

Cell phone: +420 774 784 460

Office: +420 467 070 339

Contipro a.s.

561 02 Dolni Dobrouc 401

www.contipro.com

VAT 60917431

Reviewer 2 Report

In general, the manuscript is well written, the topic is truly interesting and innovative, and the study possesses high scientific content, pertinent references, clear self-explained figures, and the conclusion supports and resumen the main results and content of the paper. I just add some minimal comment in order to enhance the quality of the manuscript

Abstract section

Please begin your abstract with a justification or importance of your study. Please do not include too many methodology details in the abstract section, just describe the strategies you used. Details develop them well in the methodology section

Introduction section

Is clear and well-written, just relocate the objective of your work at the end of the section. Also, in this section, you can add difficulties or challenges of conventional sutures for example.

Theories and hypothesis section

This section has no references that support all your comments and affirmations. Please add enough references to support your ideas

Materials and Method section

Please add a "Materials" section and add more information about hyaluronic acid and lauroyl hyaluronan derivative (brand, CAS, MW) if you used any solvent please add to this section.

Please add the wet spinning process methodology and cite your previous publication

Author Response

Assigned Editor:          Krongkarn Watakulsin

Journal:                       Pharmaceutics

Manuscript ID:             pharmaceutics-2644409

Type:                           Article

Section:                       Biologics and Biosimilars

Special Issue:              Hyaluronic Acid for Medical Applications

Submission Received:  18 September 2023

Title:

Enhancing Dental Applications: A Novel Approach for Incorporating

Bioactive Substances into Textile Threads

First, let me thank the reviewers for their valuable and substantive comments, which are greatly appreciated by the authors of this manuscript.

Text modifications are highlighted with colors as follows:

Green – new text,

Yellow – removed text,

Gray – replaced text.

Reviewer 2

Comments and Suggestions for Authors

Reviewer: In general, the manuscript is well written, the topic is truly interesting and innovative, and the study possesses high scientific content, pertinent references, clear self-explained figures, and the conclusion supports and resumen the main results and content of the paper. I just add some minimal comment in order to enhance the quality of the manuscript.

Answer: We are very grateful and appreciate this very positive evaluation of the manuscript.

Abstract section

Reviewer: Please begin your abstract with a justification or importance of your study. Please do not include too many methodology details in the abstract section, just describe the strategies you used. Details develop them well in the methodology section.

Answer: Thank you for this recommendation. The abstract has been revised according to this comment.

Introduction section

Reviewer: Is clear and well-written, just relocate the objective of your work at the end of the section. Also, in this section, you can add difficulties or challenges of conventional sutures for example.

Answer: Thank you for this comment. Whole paragraphs have been added to the Introduction section. The Introduction section has been modified and extended. And while we understand your recommendation, this section is now interspersed with general information and parts of our focus. In our opinion, the relocation of the paragraph in your recommendation would not improve the clarity of this section further.

Theories and hypothesis section

Reviewer: This section has no references that support all your comments and affirmations. Please add enough references to support your ideas.

Answer: Thank you for this comment, which is perfectly obvious. Two references used earlier in the text and one new one have been added to this section. The small number of references in this section is due to the fact that these theoretical findings are based on our experimental research and have no theoretical basis in the literature.

Materials and Method section

Reviewer: Please add a "Materials" section and add more information about hyaluronic acid and lauroyl hyaluronan derivative (brand, CAS, MW) if you used any solvent please add to this section.

Answer: Thank you for this notice. Paragraph 3.1 Materials has been added to the manuscript and a section on thread preparation has been extended.

Reviewer: Please add the wet spinning process methodology and cite your previous publication.

Answer: The description of the wet spinning method is quite extensive and beyond the scope of this paper. Therefore, we refer here to the previous work of our colleagues in references [5] and [25], as already mentioned in the original manuscript.

Sincerely yours,

Marek Pokorny

First and corresponding author

Marek Pokorny, Ph.D.

Senior Researcher / Head of the Group

Group of R&D Technologies

Cell phone: +420 774 784 460

Office: +420 467 070 339

Contipro a.s.

561 02 Dolni Dobrouc 401

www.contipro.com

VAT 60917431

Round 2

Reviewer 1 Report

Thank you for submitting the revised manuscript and for the diligent effort you have made in addressing the reviewer's comments. I appreciate the time and effort you have devoted to revising the manuscript.